# High Antitumor Activity of the Dual Topoisomerase Inhibitor P8-D6 in Breast Cancer

**DOI:** 10.3390/cancers14010002

**Published:** 2021-12-21

**Authors:** Inken Flörkemeier, Tamara N. Steinhauer, Nina Hedemann, Jörg Paul Weimer, Christoph Rogmans, Marion T. van Mackelenbergh, Nicolai Maass, Bernd Clement, Dirk O. Bauerschlag

**Affiliations:** 1Department of Gynaecology and Obstetrics, Christian-Albrechts-University Kiel and University Medical Center Schleswig-Holstein Campus Kiel, 24105 Kiel, Germany; inken.floerkemeier@uksh.de (I.F.); Nina.Hedemann@uksh.de (N.H.); Joerg-Paul.Weimer@uksh.de (J.P.W.); Christoph.Rogmans@uksh.de (C.R.); MarionTina.vanMackelenbergh@uksh.de (M.T.v.M.); Nicolai.Maass@uksh.de (N.M.); 2Pharmaceutical Institute, Department of Pharmaceutical and Medicinal Chemistry, Christian-Albrechts-University Kiel, 24118 Kiel, Germany; tsteinhauer@pharmazie.uni-kiel.de (T.N.S.); bclement@pharmazie.uni-kiel.de (B.C.)

**Keywords:** breast cancer, drug development, dual topoisomerase inhibitor, apoptosis, 2D, 3D

## Abstract

**Simple Summary:**

Despite significant advancement in therapeutic strategies, breast cancers remain the most prevalent type of cancer in terms of incidence and mortality worldwide. Therefore, new alternative therapies have become an urgent clinical need. The present study shows a significantly enhanced cytotoxicity and apoptosis of a treatment with the dual topoisomerase inhibitor P8-D6 compared to standard cytostatics. These effects were visible in various breast cancer cell lines and primary patient cells in 2D monolayer and 3D spheroids. In summary, P8-D6 seems to be a potent chemotherapeutic agent for various breast cancer cell treatments.

**Abstract:**

Breast cancer constitutes the leading cause of cancer deaths among females. However, numerous shortcomings, including low bioavailability, resistance and significant side effects, are responsible for insufficient treatment. The ultimate goal, therefore, is to improve the success rates and, thus, the range available treatment options for breast cancer. Consequently, the identification, development and evaluation of potential novel drugs such as P8-D6 with seminal antitumor capacities have a high clinical need. P8-D6 effectively induces apoptosis by acting as a dual topoisomerase I/II inhibitor. This study provides an overview of the effectiveness of P8-D6 in breast cancer with both 2D monolayers and 3D spheroids compared to standard therapeutic agents. For this drug effectiveness review, cell lines and ex vivo primary cells were used and cytotoxicity, apoptosis rates and membrane integrity were examined. This study provides evidence for a significant P8-D6-induced increase in apoptosis and cytotoxicity in breast cancer cells compared to the efficacy of standard therapeutic drugs. To sum up, P8-D6 is a fast and powerful inductor of apoptosis and might become a new and suitable therapeutic option for breast cancer in the future.

## 1. Introduction

Breast cancer (BC) is the most lethal malignancy diagnosed in women worldwide [1], given that 13% of women will develop BC in their lifetime and 15% will die [2]. Even though BC incidence has increased in recent years, the mortality rate has decreased, mainly due to earlier diagnosis and better treatment options. The major cause of chemotherapy failure in BC treatment is chemoresistance. A variety of mechanisms to avoid the cytotoxic effects of drugs can be activated in cancer cells, e.g., decreased influx or increased efflux of drugs, activated survival pathways, or enhanced DNA repair mechanisms [3]. Thus, the development of new drugs is highly warranted.

The new anticancer drug P8-D6, an aza-analogous Benzo[c]phenanthridine, was synthesized with a new simple and optimized four-step approach [4]. Moreover, P8-D6 appeared as an extremely suitable anticancer drug candidate, due to its physicochemical properties and strong cytotoxic activities. Its cytotoxicity is induced by dual topoisomerase (topo) I/II inhibition. Thereby, P8-D6 functions as a topo poison and covalently stabilizes the DNA-topoisomerase complex of both topo enzymes I and II [4,5]. Human DNA topos are essential modulators of DNA topology as these enzymes regulate the untangling and unwinding of DNA strands during cellular processes such as DNA transcription, replication or recombination. Functionally, topo I acts as single-strand DNA endonuclease and ligase that functions mainly during transcription and DNA replication [6], which enables controlled rotation of the DNA strands. Topo II, with its homologous isoforms α and β, is responsible for the double-strand break to remove DNA supercoils and DNA replication intertwining [7,8]. Due to the extension of the DNA strand breaks through the stabilization of the topo-DNA-intermediate by P8-D6, cell death is initiated by inducing apoptosis [9]. This leads to a reduction in tumor cell survival (Figure 1).

The evaluation of the effectiveness of P8-D6 in cancer was tested by the National Cancer Institute, Maryland [10]. The NCI-60 DTP Human Tumor Cell Line screening resulted in an average growth inhibition of 50 % (GI_50_) over all 60 human tumor cell lines of 49 nM [4]. For BC cells, the result showed a GI_50_ value of 0.13 µM compared to cisplatin with a GI_50_ of 23.55 µM, etoposide with a GI_50_ of 1.39 µM or epirubicin with a GI_50_ of 0.14 µM, respectively [10]. Thus, P8-D6 proved to be highly active compared to current standard therapeutics.

Cell-based two-dimensional (2D)- monolayer cell culture assays are an effective and established approach for initial drug testing [11], but three-dimensional (3D) structure models provide a more complex network of cell–matrix and cell–cell interactions, simulating the function of biological complexity more closely to in vivo settings [12,13,14].

Continuing the development of novel drugs with higher efficacy, lower resistance potential and fewer side effects is still a clinical need in BC therapy. In order to create new therapeutic options, the targeted, cell-based preclinical testing of new active drugs is essential and forms the basis of drug development. The present study outlines a strong antitumor effect of the dual topo inhibitor P8-D6 in BC.

## 2. Materials and Methods

P8-D6 was synthesized as previously described [4] and solved in PBS. Topotecan, etoposide, cisplatin and epirubicin were obtained from the UKSH dispensary.

### 2.1. Cell Preparation and Culture

The human BC cell lines MCF 7, SkBr3, MDA-MB231, MDA-MB468 and BT-20 were obtained from ATCC. Cells were maintained in RPMI 1640 medium supplemented with 10% fetal bovine serum (FBS) and 60 IU (µg)/mL penicillin–streptomycin. Primary BC (UF-182) cells were isolated from advanced stage breast cancer patients during surgery at first diagnosis (UKSH, Campus Kiel). The tumor cells were extracted from tumor tissue as described previously [15] and cultured in supplemented RPMI-1640 medium. Cells were incubated in standard 5% CO_2_ humidified incubators at 37 °C. Cells were authenticated by Short Tandem Repeat (STR) DNA profiling analysis before [16] and during culturing and routinely checked for mycoplasma contamination using MycoAlert™ (Lonza, Basel, Switzerland). Tissue donors signed informed consent agreements under the ethical approval of the Regional Ethical Review Board of the University Medical Center Schleswig–Holstein (AZ: D578/20).

### 2.2. Western Blot

Cellular protein was harvested and protein contents were determined as previously described [17]. The same amounts of protein for each sample were loaded and separated by SDS-PAGE (sodium dodecyl sulphate–polyacrylamide gel electrophoresis). Separated proteins were transferred to a PVDF (Polyvinylidene fluoride) membrane (Hybond-P, Amersham GE Healthcare, Boston, MA, USA), blocked in TBST and 5 % milk powder, and incubated with primary antibodies (anti-Topo I 1:500 (Santa Cruz Biotechnology, #sc-271285, (Heidelberg, Germany)), anti-Topo II α/β 1:10000 (Abcam#ab109524, (Cambridge, UK)) and anti-HSP 90 1:10000 (Santa Cruz Biotechnology, #sc-13119, (Heidelberg, Germany)). Incubation with HRP-labeled anti-mouse IgG 1:2000 (Santa Cruz Biotechnology, #sc-516102 (Heidelberg, Germany)) or HRP-labeled goat anti-rabbit IgG 1:3000 (Elabscience #E-AB-1003 (Houston, TX, USA) followed. After washing with TBST, membranes were developed using the ECL Plus Western Blotting Detection System (GE Healthcare). HSP 90 was used as loading control. Chemiluminescence was visualized using the ChemoStar Touch ECL & Fluorescence Imager (Intas Science Imaging Instruments, Göttingen, Germany).

### 2.3. Fluorescence Imaging

Due to its chemical structure, P8-D6 has fluorescent properties (462_Ex_/530_Em_). MCF-7 (15,000/well) and SkBr3 (20,000/well) cells were seeded in glass-bottomed 8-well chamber slides. Subsequently, cells were treated with 10 µM P8-D6 or PBS in Live Cell Imaging Solution (LCIS) with 10% FBS for 10 h, washed with LCIS. Samples were directly microscopically examined to detect the localization of P8-D6. Subsequently, samples were fixed with acetone, blocked with goat-serum (1:20 in PBS) and stained with primary antibodies (anti-Topo I 1:50 (Santa Cruz Biotechnology, #sc-271285, (Heidelberg, Germany))) and secondary antibody (Goat anti-Mouse Alexa Fluor 594 (Thermo Fisher A-11032, (Waltham, MA, USA))) plus DAPI/mounting medium (0.5 µg/mL) (Vectashield, (Burlingame, CA, USA)). The other samples, which were incubated with P8-D6 and washed with LCIS, were stained using CellTracker^TM^ Deep Red Dye (5 µM at 37 °C for 15 min) (Thermofisher Scientific, Bedford, MA, USA) and Hoechst 33342 (0.005 %) for 20min in LCIS. Then, they were washed with LCIS and fluorescence imaging was performed using the Zeiss LSM 880 microscope (Carl Zeiss Microscopy, Jena, Germany); the ZEN 2.5 software (blue edition) was used.

### 2.4. Two-Dimensional Viability and Apoptosis Assay

For the measurements of viability and apoptosis using the ApoLive-Glo™ Multiplex Assay kit (Promega #G6410, (Walldorf, Germany)), MCF-7 (40,000/well), SkBr3 (20,000/well), MDA-MB231 (40,000/well), MDA-MB468 (10,000/well), BT-20 (19,250/well) and UF-182 (7500/well) cells were seeded in a 96-well plate (Corning #3903, (Wiesbaden, Germany)). Cell lines and patient-derived cells were treated as triplicates with control and treatment groups for 48 h. The measurement was performed as described in the instructions provided by the manufacturer (TM325). Viability was measured in fluorescence units (400Ex/505Em) after 30 min (RT). Immediately afterwards, Caspase-Glo 3/7 reagent was added to each well, incubation took place for 1 h (RT) and luminescence units were measured using a microplate reader (Infinite 200, Tecan). The viability outcomes were used to normalize the caspase results (relative caspase activity: caspase activity divided by the viability (normalized to control)). With the viability data, dose–response curves were plotted using four parameter logistic regressions and IC_50_ values were calculated (GraphPad Software, Inc., San Diego, CA, USA).

### 2.5. Three-Dimensional Cytotoxicity, Viability and Apoptosis Assay

MCF-7 (1000/well), SkBr3 (1000/well), MDA-MB231 (2500/well), MDA-MB468 (500/well), BT-20 (18,750/well) and UF-182 (20,000/well) cells were seeded onto a 96-well Ultra-Low Attachment plate (Corning #4520). The number of cells for seeding was determined experimentally. The aim was to create a spheroid with a size of 400–600 μm after 96 h. To form spheroids, cells were maintained for 96 h, then half of the medium was removed and treated as triplicates with control and treatment groups for 48 h. Simultaneously, CellTox™ Green assay (Promega #G8731) was added and detected (485Ex/520Em) 24 h and 48 h after treatment using NYONE^®®^ Scientific (SYNENTEC) with 4× magnification [18]. The following excitation sources and emission filters were used: Brightfield—BFEx/GreenEm (530/43 nm); CellToxTM Green (BlueEx (475/28 nm)/GreenEm (530/43 nm)). Subsequently, the viability and apoptosis of the cells were determined by means of the RealTime-Glo™ Metabolic Cell Viability (460Em) (Promega #G9711) and Caspase-Glo 3/7 (565Em) (Promega # G8090) luminescence assays. Using a microplate reader (Infinite 200, Tecan, Blue 2NB filter, RedNB filter) with filters, both parameters were separately detectable. The first medium was removed to a rest volume of 25 µL, and 25 µL RealTime-Glo™ were added followed by incubation (1 h, 37 °C). After luminescence measurement, the samples were incubated with 25 µL Caspase-Glo 3/7 (1 h, RT) and measured again. The viability outcomes were used to normalize the caspase results (relative caspase activity: caspase activity divided by the viability (normalized to control)).

For staining with propidium iodide (10 µg/mL), calcein-AM (1 mM) and Hoechst 33342 (0.001%), the cells were grown and treated for 48 h as described above. Then, 80 % of the medium was removed and transferred with staining mixture for 3 h. For imaging, the NYONE^®®^ Scientific (SYNENTEC) microscope was used, with 4× magnification and with the following excitation sources and emission filters: Brightfield—BFEx/GreenEm (530/43 nm); Hoechst 33342—UVEx (377/50 nm)/BlueEm (452/45 nm); calcein-AM—BlueEx (475/28 nm)/GreenEm (530/43 nm); propidium iodide—LimeEx (562/40 nm)/RedEm (628/32 nm).

### 2.6. Scanning Electron Microscopy (SEM)

Spheroids were grown as described in section Three-Dimensional Cytotoxicity, Viability and Apoptosis Assay. The spheroids were treated with 1 µM P8-D6 and PBS for 48 h. Then, the spheroids were fixed with 2.5% glutaraldehyde (1 h RT). After washing, the second fixation was performed using 1% osmium tetroxide (1.5 h RT). Before dehydration using ethanol (25, 50, 75, 96, 2 × 100%, 30 min each), the spheroids were washed with PBS. Air drying using hexamethyldisilazane was performed on charcoal stubs overnight. For better conductivity, the spheroids were coated with gold and then measured with SEM (Phenom XL, Phenom-world).

### 2.7. Statistical Analysis

Frequency distributions and statistical tests were performed using GraphPad Prism 9 (GraphPad Software version 9.1.1). The means of at least three replicates were calculated. First, Gaussian distributions of the means for each treatment were tested using the Shapiro–Wilk normality test. Parametric data of multiple groups were analyzed with one-way analysis of variance (one-way ANOVA) for statistical significance. Non-parametrical datasets of multiple groups were analyzed with the Friedman test. Statistically significant differences between the groups were assumed at *p*-values < 0.05 according to Tukey’s multiple comparison test (parametric data) and Dunn’s method (non-parametric data), respectively. Statistically significant differences with *p*-values < 0.05 were marked with an asterisk (*).

## 3. Results

The process of drug development is complex. In addition to the important aspects of target identification and target reach, which are key factors for drug effectiveness, the safety profile is another essential pillar of drug development. To investigate the toxicity of P8-D6, we determined the hepatotoxicity of P8-D6 on human hepatocytes in previous studies and tested P8-D6 on non-tumor-associated ovarian epithelia cells. In that analysis, the cell integrity in non-cancer cells was slightly affected, and no hepatotoxic effect was detected in in vitro studies [19]. In characterizing the mechanism of action of P8-D6, it was previously shown that it acts as a dual topo inhibitor [5]. A major requirement for drug effectiveness is that the drug reaches its nuclear target structure-DNA-topo I and II complex. In addition to previous studies on colon cancer and ovarian cancer cells [5,19], this study confirms the localization of P8-D6 in the cytoplasm and the nucleus of BC cells after treatment (Figure 2A–C). It was determined that P8-D6 reached the cell nucleus by quantifying the fluorescence intensity per nuclear area (Figure 2C, Appendix A) In addition, the colocalization of P8-D6 and topo I was detected by fluorescence staining (Figure 2(Aii), Appendix A).

Additionally, the expression pattern of topos in BC cells was verified by western blots. High topo I expression was detected across the BC cells. In contrast, an excessive expression of topo II was mainly determined in SkBr3 und MCF-7 cells (Figure 2D and Appendix A).

### 3.1. P8-D6 Is Highly Effective in BC 2D Monolayers

The main aim of this study was to prove the efficacy of P8-D6 for BC therapy. Firstly, BC were seeded as 2D monolayers and treated with P8-D6, and compared to treatment with topotecan, etoposide and cisplatin. Thereby, this study analyzed a variety of different BC cell types with different expression levels of estrogen- and progesterone-receptor and Her2. MCF-7 is a luminal BC type, and SkBr3 have high Her2 expression but are negative for hormone receptors. MDA-MB231, MDA-MB468, BT-20 and UF-182 belong to triple-negative breast cancer (TNBC). Samples were analyzed by determining apoptosis based on Caspase 3/7 activity. The validation of these methods was conducted using a second method based on flow cytometry and Annexin V/7AAD staining as a complementary method in our previous study [19]. As these two methods delivered similar results, the caspase 3/7 assays were used for further studies. Moreover, this method has the advantage that it is performable in a plate-based formats and allows a direct and robust comparison between 2D and 3D set-ups. P8-D6 led to a significantly higher rate of caspase 3/7 activity in all tested BC cells after 48 h treatment when compared to standard therapeutics (Figure 3A–E). Even with MCF-7, which showed only marginal caspase-3 expression [20], a significant increase in caspase-7 was detected in cells treated with P8-D6 versus PBS or standard chemotherapeutics. Strikingly P8-D6 was highly active in primary BC cells established from tumor samples. (Figure 3F). Pronounced antiproliferative changes by P8-D6 treatment were also observed using microscopy imaging (Figure 3G). To calculate IC_50_ values, the viability of different BC cells after 48 h treatment was measured. Interestingly, BT-20 cells and MDA-MB468 cells had notably lower IC_50_ values compared to MDA-MB231, all of which are TNBC cells (Figure 3H). Correlating the topo II expression levels with the IC_50_ values indicated that an overexpression of topo II slightly improved the response of P8-D6. Altogether, P8-D6 induced a strong antitumor effect in all BC cells in the 2D monolayer.

### 3.2. P8-D6 Induces Strong Effects in 3D Target Tumor

Three-dimensional cell culture drug penetration, drug gradients and altered gene expression more closely resemble the in vivo cell environment by mimicking cell–cell interaction. Therefore, BC spheroids were grown in ULA plates for 96 h and subsequently treated. The spheroid morphology and growth behavior of BC cell lines differed fundamentally from the compact ex vivo primary cells (Figure 4A–F). After treatment with P8-D6, a slight change in the growth behavior and, more clearly, a change in the intracellular stability of the spheroids could be detected.

Furthermore, the relative caspase 3/7 activity was significantly increased in all P8-D6 treated spheroids (MCF-7, SkBr3, MDA-MB231, MDA-MB468, BT-20 and UF-182) compared to topotecan treatment and negative control [PBS] (Figure 5A–F). Topotecan induced an increased apoptosis rate, especially in cell lines and in UF-182 cells, compared to the negative control, similarly to the results obtained in the 2D monolayer experiments. However, topotecan was not as effective as P8-D6 in BC spheroids. Using live-dead staining followed by automated fluorescence and bright field microscopy confirmed that P8-D6 is a highly active compound compared to topotecan and PBS. It was clearly determined that for P8-D6 treatment compared to references, the number of dead cells increased and the number of viable cells and cell nuclei decreased. (Figure 6A–C). Cell toxicity was quantified during treatment by measuring the fluorescence signal of CellTox™ Green (Figure 6D,E). P8-D6 treatment, especially with the 10 µM concentration, induced a visually and quantitatively observable toxic effect in BC spheroids (MCF-7, SkBr3, MDA-MB231, MDA-MB468, BT-20 and UF-182). To characterize and visualize the surface chances of SkBr3 and UF-182 spheroids after treatment, scanning electron microscopy (SEM) was Houston, Texas (Figure 6F). Treatment with P8-D6 caused a decrease in membrane integrity in the SkBr3 and UF-182 spheroids’ surfaces, and thus, the spheroids became more porous in their structure.

## 4. Discussion

BC is currently one of the deadliest cancers [21]. For this reason, a suitable and successful therapy is essential. In addition to surgery and radiation therapy, systemic therapy is a further important pillar in BC therapy. However, the death rate and the development of metastasis show that BC treatment is not fully successful at present. These effects are favoured due to chemotherapy resistance and treatment limitations because of side effects [22]. Consequently, the development of new, active therapeutic drugs such as P8-D6 has a high clinical need.

In recent years, it became evident that there are several subtypes of breast cancer such as luminal A type, Her2 type, and the triple-negative breast cancers. These subtypes are of different aggressiveness levels and are treated in different fashions. To prove that the newly developed dual topoisomerase inhibitor P8-D6 is effective across all subtypes, we investigated several well-characterized breast cancer cell lines. This also addressed the known tumor heterogeneity of breast cancer.

The induction of apoptosis in BC cells was significantly higher by P8-D6 treatment compared to standard therapeutics and negative control. The present study clearly determined the high antitumor property of P8-D6 in molecularly different BC cell lines and ex vivo primary patient cells in 2D monolayers and 3D culture. Besides, the 2D-Monolayer was often used in the past to investigate drug effects. The disadvantage of this model is the limited predictability of anticancer drug efficacy due to the absence of tissue structure and architecture, which influences both pharmacokinetics and pharmacodynamics. Enhanced, reliable options to replace 2D monolayers include 3D culture models, which contribute considerably to improving the expressiveness and validity of in vitro preclinical models in anti-cancer drug testing.

The development of resistances during chemotherapy contributes to therapy failure. The inhibition of both topoisomerases (topo I/II) can prevent the development of resistance, since when only one of the two enzymes is inhibited, the other is upregulated [23,24]. In addition to the high effectiveness, this is an essential advantage of dual topo inhibition. P8-D6 acts as such a dual topo I/II inhibition.

Topo inhibitors play an important role in breast cancer treatment. Topo II inhibitors, such as doxorubicin, are recognized as highly active drugs for breast cancer treatment, despite their dose-dependent cardiac toxicity [24,25]. New liposomal formulations decrease the side effect profile with similar efficacy [26,27]. Topo I inhibitors are used less frequently in the treatment of metastatic breast cancer. The development of resistances during chemotherapy is the main reason for therapy failure. The inhibition of both topoisomerases (topo I/II) can prevent the development of resistance, since when only one of the two enzymes is inhibited, the other is upregulated [23,28]. In addition to the high effectiveness, this is an essential advantage of dual topo inhibition. P8-D6 acts as such a dual topo I/II inhibitor.

Currently, topo inhibitors are being tested in clinical trials. Irinotecan is a prodrug of the active metabolite SN-38, acting as a DNA topo I poison [29]. It is presently being tested in a phase-2 clinical trial as a nanoliposomal drug for metastatic breast cancer [30]. Several studies on the free active compound irinotecan have already been completed [31,32]. No dual topo inhibitor is currently approved for either BC or other entities. The dual topo inhibitor TAS-103 has been tested in clinical trials, but has not been approved yet [33]. This shows the novel mode of action of P8-D6. All these findings illustrate that topoisomerase inhibitors are already increasingly used in breast cancer therapy. Our study demonstrates the overarching effectiveness of dual topoisomerase inhibition against multiple different types of breast cancer. Due to its high cytotoxic potential on cancer cells, P8-D6 even improves the treatment of poorly responding TNBC. Moreover, the combination therapy of P8-D6 with hormone-based drugs or treatment targeted to Her2 may especially improve luminal-type BC or Her2-positive BC outcomes. For this reason, the combination therapy with P8-D6 and trastuzumab or tamoxifen should be examined more closely.

## 5. Conclusions

Breast cancer remains a challenging cancer type to treat. This present study shows the outstanding apoptotic effect of P8-D6 in BC cell lines and in a translational approach in ex vivo BC primary patient cells, in both 2D monolayers and 3D culture, compared to standard therapeutics. In order to prove the benefit of P8-D6 treatment for BC therapy in multiorgan systems and to verify potential toxic or side effects, further in vivo experiments would be beneficial.

## Figures and Tables

**Figure 1 cancers-14-00002-f001:**
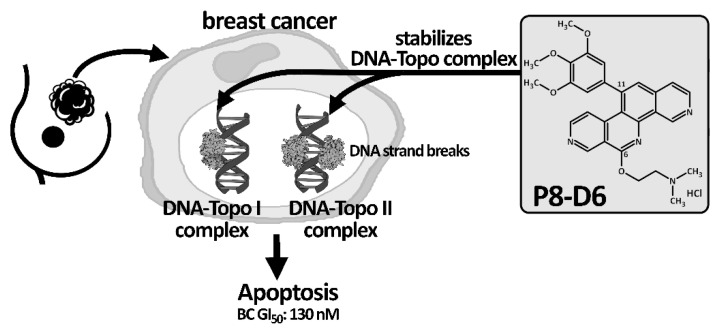
Schematic mechanism action of P8-D6. As a topo I/II poison, P8-D6 covalently stabilizes the enzyme–DNA complex and, thus, increases the amount of single- and double-strand DNA breaks and subsequently causes cell death. Initially, effectiveness and the broad activity spectrum of P8-D6 was verified in the NCI-60 DTP human tumour cell line screening by the NCI. P8-D6 was able to induce a 50 % growth inhibition in 60 cell lines in nanomolar concentrations (49 nM) [4]. In BC cells, P8-D6 reached an average GI_50_ (BC) value of 130 nM. GI_50_ (BC): average growth inhibition of 50% in BC cell lines.

**Figure 2 cancers-14-00002-f002:**
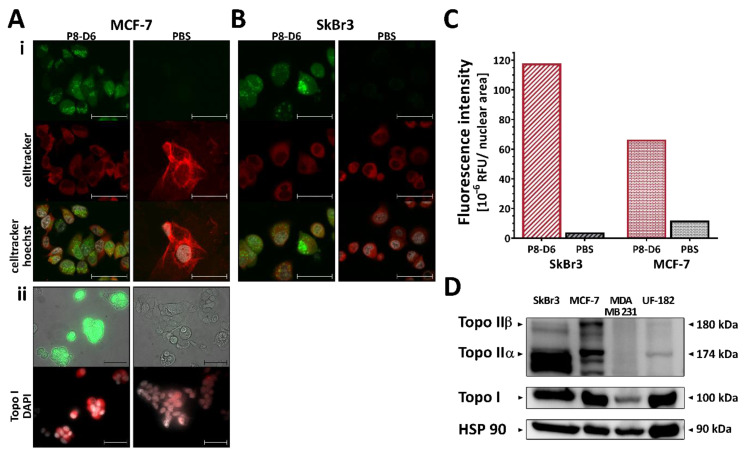
Target control. MCF-7 (**A**) and SkBr3 (**B**) cells were treated with 10 µM P8-D6 (fluorophore: 462Ex/530Em) or control (PBS) for 10 h. P8-D6 were localized in vitro. (**i**): Cells were stained by CellTracker^TM^ Deep Red Dye and Hoechst 33342. (**ii**) After fixation, topoisomerase I were detected. Fluorescence images show the fluorophore P8-D6 in green, membrane staining (**i**) or topoisomerase expression (**ii**) in red and nucleus staining in white. Scale bars, 50 µm. (**C**) Fluorescence intensity of P8-D6 in the nuclei was compared to PBS control. (**D**) BC cell lines and primary cells were lysed and protein expression was analyzed using western blot. HSP 90 was used as loading control.

**Figure 3 cancers-14-00002-f003:**
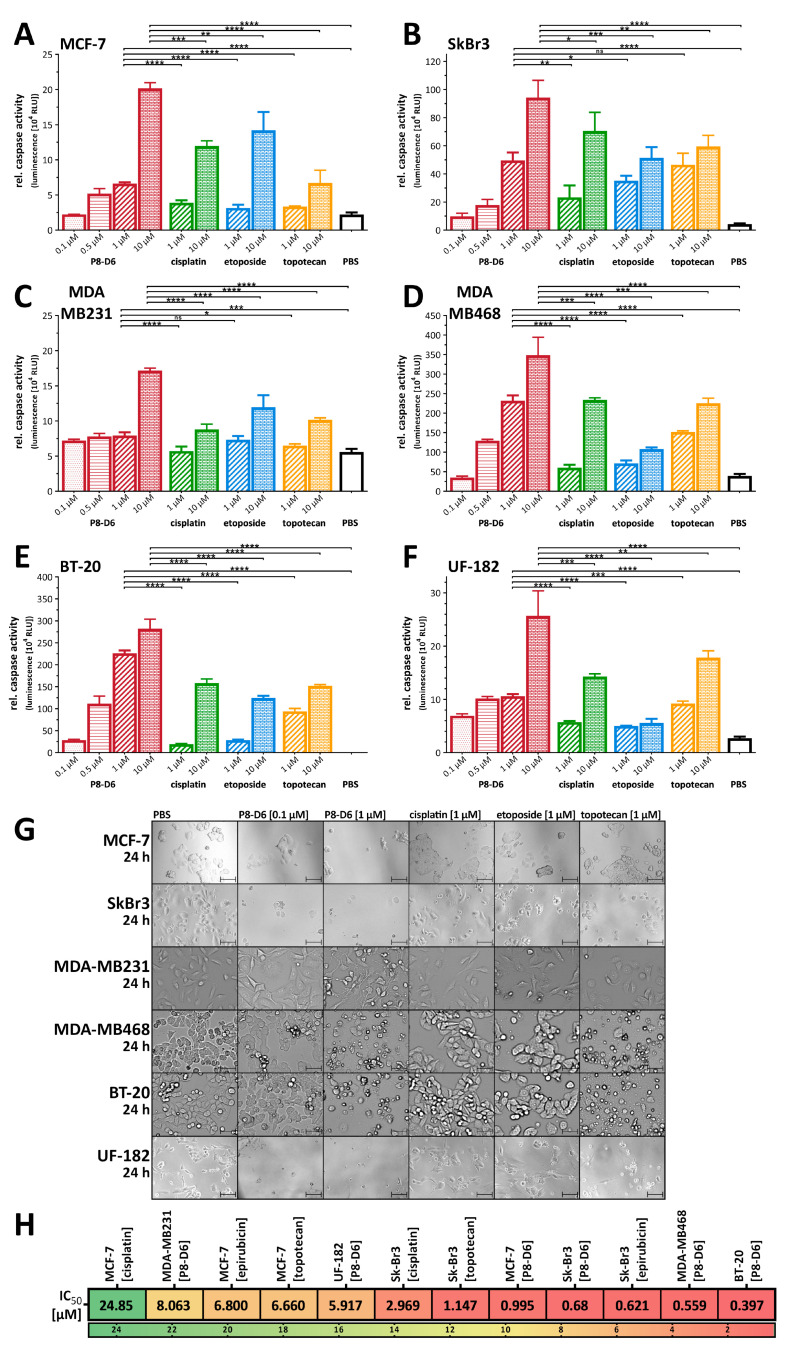
Antitumor responses in BC 2D monolayers. MCF-7, SkBr3, MDA-MB231, MDA-MB468, BT-20 (cell line) and UF-182 (primary cells) were treated with P8-D6, cisplatin, etoposide, topotecan and PBS as control. The relative caspase activity representing the rate of apoptosis were measured 48 h after treatment in MCF-7 (**A**), SkBr3 (**B**), MDA-MB231 (**C**), MDA-MB468 (**D**), BT-20 (**E**) and UF-182 (**F**) cells. Additionally, the anti-proliferative effects in the cells were visualized by microscopy after 24 h treatment. Scale bars, 50 µm (**G**). Heat map presents the IC50 values calculated by viability (**H**). P8-D6 was compared to cisplatin, epirubicin and topotecan. Data are means + SD one-way ANOVA, * (*p* < 0.05), ** (*p* < 0.01), *** (*p* < 0.001), **** (*p* < 0.0001), ns (non-significant).

**Figure 4 cancers-14-00002-f004:**
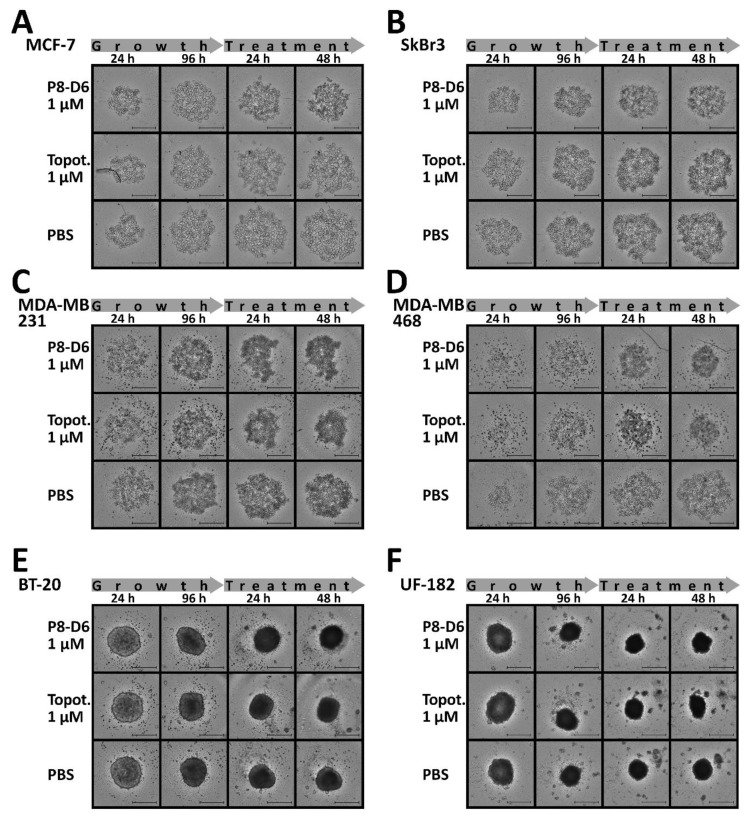
Growth changes in BC spheroids. For 3D culture, MCF-7, SkBr3, MDA-MB231, MDA-MB468, BT-20 (cell line) and UF-182 (primary cells) cells were cultured in ULA plates for 96 h. Subsequently, spheroids were treated with P8-D6 (10 µM, 1 µM, 0.5 µM, 0.1 µM), topotecan (10 µM, 1 µM) and PBS for 48 h. For the monitoring of growth changes and morphological changes, spheroids were imaged every 24 h by microscopy. MCF-7 (**A**), SkBr3 (**B**), MDA-MB231 (**C**), MDA-MB468 (**D**), BT-20 (**E**) and UF-182 (**F**). Scale bars, 500 µm.

**Figure 5 cancers-14-00002-f005:**
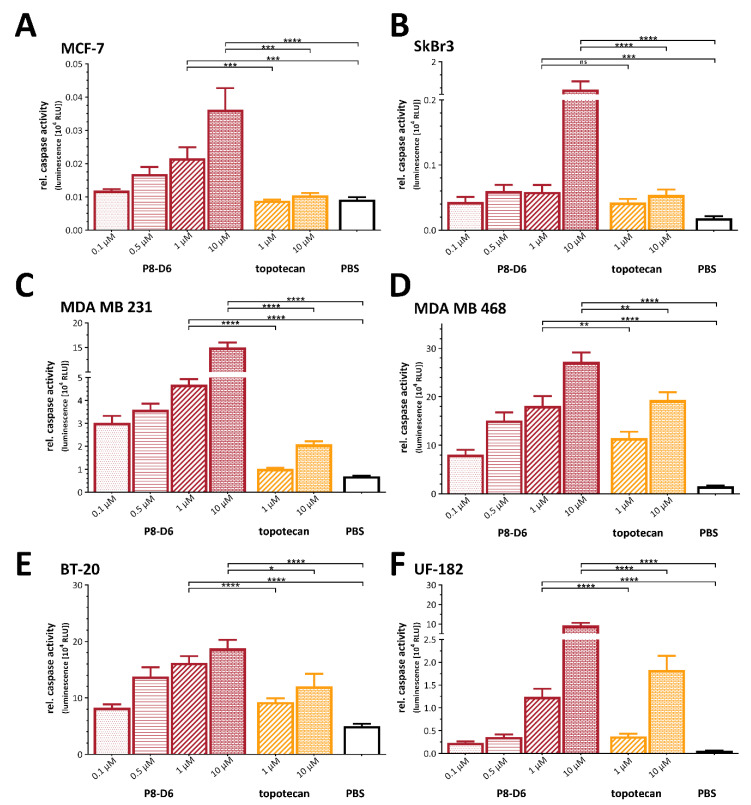
Apoptosis induction in BC spheroids. For 3D culture, MCF-7, SkBr3, MDA-MB231, MDA-MB468, BT-20 (cell line) and UF-182 (primary cells) cells were cultured in ULA plates for 96 h. Subsequently, spheroids were treated with P8-D6 (10 µM, 1 µM, 0.5 µM, 0.1 µM), topotecan (10 µM, 1 µM) and PBS for 48 h. After treatment, the viability and caspase activity were analyzed in MCF-7 (**A**), SkBr3 (**B**), MDA-MB231 (**C**), MDA-MB468 (**D**), BT-20 (**E**) and UF-182 (**F**) spheroids. Data are means + SD (*n* = 3) one-way ANOVA, * (*p* < 0.05), ** (*p* < 0.01), *** (*p* < 0.001), **** (*p* < 0.0001), ns (non-significant).

**Figure 6 cancers-14-00002-f006:**
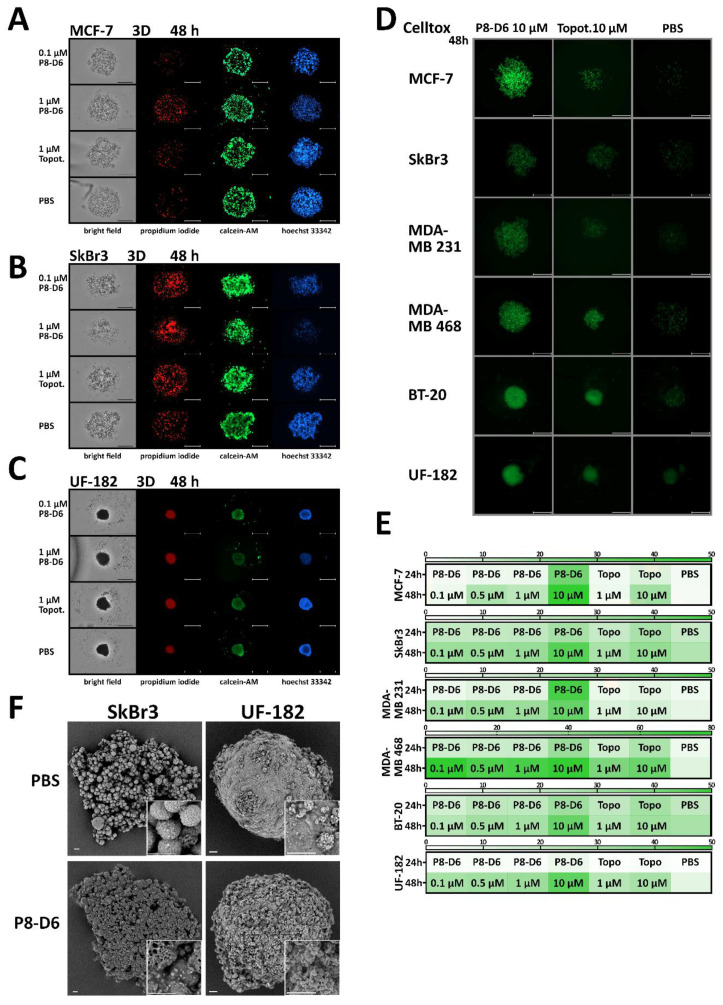
Cell toxicity, live-dead staining and morphological changes in BC spheroids. For 3D culture, MCF-7, SkBr3, MDA-MB231, MDA-MB468, BT-20 (cell line) and UF-182 (primary cells) cells were cultured in ULA plates for 96 h. Subsequently, spheroids were treated with P8-D6 (10 µM, 1 µM, 0.5 µM, 0.1 µM), topotecan (10 µM, 1 µM) and PBS for 48 h. After treatment, the spheroids were stained with propidium iodide (red), calcein-AM (green) and Hoechst 33342 (blue) and imaged using NYONE^®®^ Scientific. MCF-7 (**A**), SkBr3 (**B**) and UF-182 (**C**). Scale bars, 500 µm. (**D**) During treatment, the cell toxicity was measured by fluorescence microscopy using CellTox™ Green (24 h, 48 h). Scale bars, 500 µm. These fluorescence signals were quantified (fluorescence intensity RFU) and presented in a heat map (**E**). (**F**) The P8-D6- (1 μM) or PBS-treated SkBr3 and UF-182 spheroids (48 h) were analyzed by scanning electron microscopy (SEM). Scale bars, 20 µm.

## Data Availability

The data presented in this study are available on request from the corresponding author.

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
