# Peer review of "High Antitumor Activity of the Dual Topoisomerase Inhibitor P8-D6 in Breast Cancer"

_cancers, 2021, doi:10.3390/cancers14010002_

Round 1

Reviewer 1 Report

Another method need to be used for apoptosis detection.

Without any mechanistical study data and in vivo data, the authors at least need to show whether modulation ( for example knock down of Top I or II) of TopI and Top II expression or the cells with different TopI and TopII expression response differently to the p8-D6.

Reviewer 2 Report

Thank you for addressing my previous comments. Minor revisions must be addressed prior to publication. 

  1. Please delete either lines 316-320 or 327-331. They are identical.
  2. Delete line 321, it is identical to line 332 and cut's of making it an incomplete sentence. 

Reviewer 3 Report

The manuscript is well revised from cancers-1405794. However, it is difficult to be understood that TOPO I was localized in the cytoplasm. Did the author detect colocalization of P8-D6/TOPO I in the nucleus? At least, it was not apparent from Figure 2A. 

Author Response

This manuscript is a resubmission of an earlier submission. The following is a list of the peer review reports and author responses from that submission.

Round 1

Reviewer 1 Report

The authors examined novel dual topoisomerase inhibitor P8-D6 in breast cancer cells and explored that P8-D6 effectively caused apoptosis in these cells.

Although the data is interesting, there are some points to be addressed.

  1. The authors stated that the localization of P8-D6 in the nucleus (line 172 and Figure 2A, B). However, from Figure 2A and B, it seems that P8-D6 is localized predominantly in the cytoplasm. It may hopeful to confirm the colocalization of P8-D6 and topo I/II by fluorescent imaging.
  2. The authors stated that P8-D6 is effective especially in HR+ or HER2+ cells than TNBC cells. However, the author used only single cell line for luminal-type and HER2-type cells (it should be noted that SKBR-3 is HR negative/HER2 positive).  Considering that chemotherapy is mainly used in TNBC patients, the authors further examine using another luminal or TNBC cell lines.
  3. MCF-7 does not express caspase 3 (doi: 10.1007/s10549-008-0217-9). Is it really OK to evaluate apoptosis using caspase 3/7 Glo for MCF-7 ?

Minor

  1. relatively poor.
  2. "monolayer 3D spheroids" ?? (line 16)

Reviewer 2 Report

The authors present a compelling story and argue that there is a need for novel therapies that target breast cancers. This manuscript demonstrates that P8-D6 a topoisomerase I/II inhibitor (stabilizer) effectively induces apoptosis of human immortalized and primary breast cancer cells in both 2D and 3D cultures. Apoptosis was measured both intrinsically and visually by various methods. Finally, P8-D6 may be a novel drug that can successfully induce apoptosis in various breast cancer cell lines and may be considered for further in vivo studies.

  1. Line 172, the authors say “this study confirms the localization of P8-D6 in the nucleus of BC cells after treatment”. However, to me there is no clear visual of nuclear staining, rather it seems to be heavily present in the membrane of both cell lines. The authors should quantify the nucleus and the membrane and compare between PBS and P8-D6 treatment.

  1. The authors also mention in the introduction that P8-D6 on average demonstrated an IC50 of 130nM specifically on breast cancer cells. Then why treat with 10uM? The authors need to explain the rationale for this.

  1. Figure 3F-G, what happen to MDA-MB-231 and SKbr3? The authors should remain consistent and show the data for all 4 cell lines throughout the entire manuscript for consistency.

  1. Figure 3A is not necessary, rather the authors should explain why they choose these cells lines and explain why TNBCs (MDA-MB-231 and UF-182) would response favorably as they replicate and proliferate much faster than ER+ MCF7 and HER2+SkBr3.

  1. Figure 4A-C. The authors should include data for MDA-MB-231.

  1. Figure 4D-F. Please explain why treatments were increased to 10uM if it’s established that 1uM is the IC50?

  1. The authors need to expand on their discussion, there is no talk about why the it’s important to test this inhibitor on multiple different types of breast cancer cell lines i.e. TNBC, ER+ and HER2+. Along with their different treatments and the potential for those treatments to be combined with P8-D6 i.e. Chemotherapy+P8-D6 or Fulvestrant, tamoxifen+P8-D6.

Reviewer 3 Report

In this manuscript, the authors demonstrated that dual topoisomerase inhibitor P8-D6 can induce apoptosis in several breast cancer cell lines using both in vitro 2D and 3D experiments. The data are convincing. However, all data only support one conclusion that P8-D6 induces apoptosis.  There are not any mechanistic studies, and the in vivo efficacy should be tested at lease in one cell line. In addition, the writing could be improved.

The manuscript in current version is not suitable to accept for publication in Cancers.